# Prospect and Challenges of Volatile Organic Compound Breath Testing in Non-Cancer Gastrointestinal Disorders

**DOI:** 10.3390/biomedicines12081815

**Published:** 2024-08-09

**Authors:** Weiyang Zheng, Ke Pang, Yiyang Min, Dong Wu

**Affiliations:** 1Department of Gastroenterology, Peking Union Medical College Hospital, Chinese Academy of Medical Sciences & Peking Union Medical College, Beijing 100730, China; weiyangzheng@aliyun.com; 2Chinese Academy of Medical Sciences & Peking Union Medical College, Beijing 100006, China; pangk19@student.pumc.edu.cn (K.P.); minyy19@mails.tsinghua.edu.cn (Y.M.); 3Clinical Epidemiology Unit, Peking Union Medical College Hospital, Chinese Academy of Medical Sciences & Peking Union Medical College, Beijing 100730, China

**Keywords:** volatile organic compounds, breath test, non-cancer gastrointestinal disorders, gastro-esophageal reflux disease, Barrett’s esophagus

## Abstract

Breath analysis, despite being an overlooked biomatrix, has a rich history in disease diagnosis. However, volatile organic compounds (VOCs) have yet to establish themselves as clinically validated biomarkers for specific diseases. As focusing solely on late-stage or malignant disease biomarkers may have limited relevance in clinical practice, the objective of this review is to explore the potential of VOC breath tests for the diagnosis of non-cancer diseases: (1) Precancerous conditions like gastro-esophageal reflux disease (GERD) and Barrett’s esophagus (BE), where breath tests can complement endoscopic screening; (2) endoluminal diseases associated with autoinflammation and dysbiosis, such as inflammatory bowel disease (IBD), irritable bowel syndrome (IBS), and coeliac disease, which currently rely on biopsy and symptom-based diagnosis; (3) chronic liver diseases like cirrhosis, hepatic encephalopathy, and non-alcoholic fatty liver disease, which lack non-invasive diagnostic tools for disease progression monitoring and prognostic assessment. A literature search was conducted through EMBASE, MEDLINE, and Cochrane databases, leading to an overview of 24 studies. The characteristics of these studies, including analytical platforms, disorder type and stage, group size, and performance evaluation parameters for diagnostic tests are discussed. Furthermore, how VOCs can be utilized as non-invasive diagnostic tools to complement existing gold standards is explored. By refining study designs, sampling procedures, and comparing VOCs in urine and blood, we can gain a deeper understanding of the metabolic pathways underlying VOCs. This will establish breath analysis as an effective non-invasive method for differential diagnosis and disease monitoring.

## 1. Introduction

Breath, often overlooked as a biomatrix, holds a significant history in the field of disease diagnosis, with its roots tracing back to the time of Hippocrates, the father of medicine (460–370 BC). Despite its early recognition, the emergence of blood tests and other imaging procedures led to a diminished focus on using exhaled breath for disease detection and diagnosis. However, in 1971, Linus Pauling, a Nobel Prize winner, revitalized modern-day breath testing by identifying the presence of 250 substances in exhaled breath. This breakthrough reignited interest in the potential of breath analysis for medical purposes, and it has been a subject of extensive research ever since [1].

Over the past five decades, both exogenous and endogenous VOCs have been investigated in numerous clinical studies, aiming to harness their potential in non-invasively diagnosing, detecting, and monitoring various human diseases, as well as evaluating the efficacy of medications [2]. Recent technological advancements have further propelled the field of breath analysis, resulting in a substantial increase in publications and clinical trials. Currently, there are an astonishing 2746 VOCs identified in humans, with 1488 of them found in exhaled breath. This expanding knowledge base has fueled the hope of leveraging breath analysis as a valuable tool in medical practice [3].

Despite the growing interest and promising developments in breath research, only a limited number of breath tests have received approval or emergency use authorization (EUA) from regulatory bodies in the United States and Europe. Among them, the urea-^13^C breath test for detecting Helicobacter pylori infection has been successfully integrated into routine clinical practice for gastrointestinal diseases [4]. However, the adoption of breath tests for other medical applications remains relatively limited.

VOCs have garnered substantial attention in breath analysis due to their organic nature and potential as biomarkers [5]. These molecules, often present within the human body, can directly reflect both normal and abnormal biological processes. As a result, they can traverse biological tissues and be exhaled, making them accessible for analysis. Various bodily fluids, including blood, exhaled air, perspiration, urine, feces, and even lacrimal fluid, can carry potential biomarkers to the exterior [6]. Once these analytes are detected, they can be accurately assessed and linked to pathological conditions or health status, thereby providing valuable information about the producing organism.

Despite the increasing number of clinical trials involving breath analysis, none of the breath tests have made a successful transition into routine clinical practice. One possible reason for this is the inherent flaw in many research designs, where most clinical trials attempt to compare the exhaled breath of healthy subjects with that of patients in advanced disease stages, aiming to identify specific breath biomarkers for disease detection [7]. However, researchers may have overlooked a crucial point: patients with any advanced disease or ailment are likely to exhibit elevated levels of inflammatory or oxidative stress-related VOCs in their breath, regardless of the specific disease etiology. Consequently, the focus on late-stage disease biomarkers may not be as relevant in real-world clinical practice.

Therefore, the objective of this review is to explore the potential application of VOC breath tests in the diagnosis of non-cancer diseases. Specifically, we will delve into three areas: (1) Precancerous conditions, such as GERD and BE, where breath tests can serve as a supplementary tool alongside endoscopic screening; (2) endoluminal diseases associated with autoinflammation and dysbiosis, including inflammatory bowel disease, irritable bowel disease, and coeliac disease, which currently rely primarily on biopsy and symptom-based diagnosis; (3) chronic liver diseases, such as cirrhosis, hepatic encephalopathy, and non-alcoholic fatty liver disease, which lack non-invasive diagnostic tools for disease progression monitoring and prognostic assessment.

By critically examining the potential of VOC breath tests in these specific disease categories, we aim to shed light on the advancements, challenges, and opportunities that lie ahead in utilizing breath analysis as a non-invasive diagnostic tool. Ultimately, our goal is to contribute to the broader understanding and practical application of breath analysis in the realm of non-cancer disease diagnosis.

## 2. Method

### Search Strategy

Literature on VOCs in non-cancer gastrointestinal diseases was searched through EMBASA, MEDLINE, and Cochrane using combinations of the following terms: “Volatile organic compound*”, “VOC”, “breath analysis”, “electronic nose”, “irritable bowel syndrome”, “inflammatory bowel disease”, “gastroesophageal reflux disease”, “barret* esophagus”, “coeliac disease”, “chronic liver disease”, “Gastrointestinal diseases”. All relevant records until 1 July 2024 written in English were included. An overview of the studies included is shown in Table 1. The electronic search was supplemented by searching published abstracts from meetings of Digestive Disease week, The Association of Surgeons of Great Britain and Ireland (ASGBI), The European Society of Surgical Research, The Association of Coloproctology for Great Britain and Ireland, The American College of Surgeons, The British Society of Gastroenterology (BSG), and the American Society of Clinical Oncology (ASCO), from 2010 to 2023. The reference lists of articles obtained were also searched to identify further relevant citations.

Abstracts of the citations identified by the search were then scrutinized to determine eligibility for inclusion in the review. Studies were included if they met the following criteria: trials analyzing endogenous VOCs from exhaled breath to diagnose or assess non-cancer gastrointestinal disease in adult humans and published after 1990 (to ensure that the methodology used for analysis was representative of current practice). Studies were excluded if an exogenous substrate (intravenous or oral) was administered prior to exhaled breath sampling, or if they were focused on cancer. Trials were excluded that analyzed VOCs from other biofluids including urine, serum, and gastric content, did not analyze VOCs from exhaled breath, and were published before 1990 (Figure 1). The quality of each study included was assessed using the quality assessment tool for diagnostic accuracy studies (QUADAS-2) [8].

**Table 1 biomedicines-12-01815-t001:** Overview of studies on VOC testing in non-cancer GI diasease.

Reference	Analytical Platform	Reference Test	Disorder Type and Stage, and Group Size	Sensitivity	Specificity	AUC	Breath Biomarkers
BE							
Peters (2020) [9]	e-nose	Endoscopy	BE (n = 129) and HC (n = 132)	91%	74%	-	-
Chan (2017) [10]	e-nose (Aeonose^TM^)	Endoscopy	BE (n = 66) and patients with suspicious history but clarified with endoscopy (n = 56)	82%	80%	0.79	-
GERD							
Dryahina (2014) [11]	SIFT-MS	Symptom-based diagnosis	GERD (n = 22) and HC (n = 24)	n.a.	n.a.	n.a.	Acetic acid (+)
IBD							
Arroyo-Manzanares (2023) [12]	GC-MS	n.a.	IBD (n = 56) and HC (n = 48)	100%	100%	n.a.	n.a.
Smolinska (2018) [13]	GC-MS	SCCAI	UC (n = 76) and non-IBD (n = 22)	92%	77%	0.94	Cumene (+), 2,4-dimethylpentane (+), methylcyclopentene (+), C_14_H_30_ branched (+) Pentadecane (−), 3-methyl-1-butanol (−), octane (−), acetic acid (−), α-pinene (−), m-cymene (−)
Dryahina (2017) [14]	SIFT-MS	HBI, SCCAI	CD (n = 136) and UC (n = 51) and HC (n = 14)	n.a.	n.a.	n.a.	Pentane, Isoprene, Hydrogen sulphide, Carboxylic acids
Arasaradnam (2016) [15]	FAIMS	HBI, SCCAI	IBD (n = 54) and HC (n = 22) CD (n = 25) and HC (n = 22) UC (n = 29) and HC (n = 22) CD (n = 25) and UC (n = 29)	74% 69% 61% 67%	75% 67% 62% 67%	0.82 0.77 0.70 0.70	n.a.
Rieder (2016) [16]	SIFT-MS	n.a.	IBD (n = 35) and patients with history or suspicious symptoms (n = 6)	n.a.	n.a.	0.81	Acetone, acrylonitrile, carbon disulfide, and triethylamine
Baranska (2016) [17]	GC-MS	Rome III	IBS (n = 170) and HC (n = 153)	89.4%	77.3%	0.83	n.a.
IBS							
Van Malderen (2022) [18]	Multicapillary Column/IMS	Rome III	IBS (n = 72) and HC (n = 24) IBS-D (n = 21) and HC (n = 24) IBS-C (n = 24) and HC (n = 24)	97% 67% 67% 92%	21% 75% 92% 50%	0.62 0.70 0.81 0.68	n.a.
Baranska (2016) [17]	GC-MS	Rome III	IBS (n = 170) and HC (n = 153)	89.4%	77.3%	0.83	n.a.
Cauchi (2014) [19]	GC-MS	Rome III	IBS (n = 28) and HC (n = 20)	41%	72%	0.44	n.a.
Patel (2014) [20]	SIFT-MS	Rome III		n.a.	n.a.	0.99	Benzene (+) Dimethyl sulfide (+) 1-octene (+) 1-3-methyhexane (+)
Chronic liver diseases							
Ferrandino (2020) [21]	GC-MS	Cirrhotic patients had an established diagnosis according to EASL and AASLD guidelines	LC (n = 32) and HC (n = 40)	n.a.	n.a.	n.a.	limonene
Vincentis (2017) [22]	e-nose (BIONOTE^TM^)	Child–Pugh Classification and MELD (A combination of clinical, biochemical, radiological, and endoscopic findings, with confirmatory liver biopsy in doubtful cases)	Child–Pugh A (n = 37), B (n = 33) and C (n = 19)	n.a.	n.a.	n.a.	n.a.
Vincentis (2016) [23]	e-nose with PLS-DA	Blood test, ultrasound	CLD (n = 104) and HC (n = 56) LC (n = 65) and NC-CLD (n = 39)	86.2% 87.5%	98.2% 69.2%	0.84 n.a.	n.a.
Pijls (2016) [24]	GC-MS	Liver biopsy and symptom	Compensated cirrhosis (n = 34) and CLD without cirrhosis (n = 87)	83%	87%	0.90	3-methylbutanal, Propanoic acid, Octane, Terpene (C10H16), Terpenoid: α-pinene, 3-carene, Branched C_16_H_34,_ 1-hexadecanol Branched C_16_H_34,_ Dimethyl disulfide
Alkhouri (2015) [25]	SIFT-MS	Liver biopsy	Advanced fibrosis (F3–4, n = 20) and CLD without advanced fibrosis (n = 41)	85%	68%	0.855	isoprene
Fernández (2015) [26]	PTR–MS	Liver biopsy	LC (n = 31) and HC (n = 30)	97%	70%	0.95	Methanol, 2-pentanone and limonene
Hanouneh (2014) [27]	SIFT-MS	Liver biopsy and etiology	AH (n = 40) and non-AH (n = 40)	80–97%	72–86%	0.92	TMA + Pentane, trimethylamine
Morisco (2013) [28]	PTR-ToF-MS	Liver biopsy	LC (n = 12) and HC (n = 14)	83%	86%	0.8869	2-butanone, 2- or 3-pentanone, C8-ketone, C9-ketone, Monoterpene, Terpene related, S-compound, Sulfoxide-compound, N-compound, Hepadienol, Methanol
Dadamio (2012) [29]	GC-MS and linear discriminant analysis	Liver biopsy	LC (n = 35) and HC (n = 49)	82–88%	96–100%	n.a.	Acetone, Styrene, Branched chain alkane, Dimethyl sulfide, Dimethylselenea, Phenola, Tetradecane, Branched chain alkane, Indolea, Octane, Isoprene, Nonane, gamma-Terpinene, 2-Methyl-1-propene, 2-Butanone, beta-Pinene, Caryophyllene

HC: healthy controls; PLS-DA: partial least square discriminant analysis; CLD: chronic liver disease; LC: liver cirrhosis, NC-CLD: non-cirrhosis chronic liver diseases; PTR-ToF-MS: proton transfer reaction time-of-flight mass spectrometry.

## 3. Results

Twenty-three studies involved patients who met the inclusion criteria and were included in this review (Table 1): three studies in GERD and BE, four studies in IBS, six studies in IBD, one study in coeliac disease, and nine studies in chronic liver diseases (Figure 2). All 23 studies evaluated were phase I biomarker studies.

### 3.1. Quality Assessment of Studies

The outcomes of the QUADAS-2 quality assessment for the included studies are illustrated in Figure 3. Over half of the studies were found to have a high risk of bias regarding patient selection, due to the lack of consecutive or random patient enrollment. Many studies also failed to avoid a case–control design. The risk of bias for the index test (the VOC breath test) was high for 75% of the studies, as most started with established cohorts of patients with known disease conditions rather than using an unselected sample. In contrast, the studies generally had a low risk of bias in the domain of reference standard as the reference tests used, such as histologic diagnosis for Barrett’s esophagus and GERD, the Rome III criteria for IBS, liver biopsy for chronic liver diseases, and the well-established Child–Pugh score for cirrhosis, are considered gold standards in clinical diagnosis. However, a few studies did not clearly specify the diagnostic standard, and were therefore assessed as having a high risk of bias in this domain. Given that the collection and analysis of VOC samples can be conveniently integrated into clinical workflows, most studies performed the VOC test concurrently with the reference standard, resulting in a low risk of bias in the flow and timing domain. Finally, as all the included studies were focused on assessing the effectiveness and efficiency of VOC breath tests, there were no concerns regarding applicability.

### 3.2. Current Analytical Platforms of VOCs

VOC analysis methods typically involve several stages: sampling, sample concentration, transfer to the analytical device, and detection and identification of individual compounds [30]. Analytical platforms can be categorized as off-line or on-line (real-time) devices [31].

Gas chromatography–mass spectrometry (GC-MS) is the most common off-line device used in VOC analysis studies. It involves vaporizing the sample and separating the mixture of gases dissolved in a mobile phase by passing it through a stationary phase. GC-MS offers high sensitivity and the ability to estimate a wide range of VOCs simultaneously. However, it has a long response time, ranging from minutes to hours. GC/MS offers high sensitivity and fast measurement processes, but it requires a trained operator; not all peaks are calibrated, and the instrumentation can be costly.

Mass spectrometry (MS) is an analytical technique that separates ionized atoms and molecules based on their mass-to-charge ratio. It allows for the identification of the chemical composition and structure of a sample. Selective ion flow tube-mass spectrometry (SIFT-MS) provides a low risk of contamination and requires short sample preparation time at high cost. Direct injection mass spectrometry (DIMS) has gained attention for the detection and quantification of VOCs in various fields such as environmental monitoring, food science and technology, and health sciences [30]. One promising DIMS method for VOC detection is proton transfer reaction-mass spectrometry (PTR-MS) [32]. In particular, the recent version of PTR-MS based on a time-of-flight (ToF) mass analyzer, known as PTR-ToF-MS, offers several advantages. It maintains ultrahigh sensitivity and low limits of detection, often in the parts per trillion by volume range. Additionally, it provides improved speed and analytical information. A single spectrum can be obtained in a split second, and in most cases, the sum formula of the observed ion peaks can be determined [33]. This makes PTR-ToF-MS a highly promising technique for VOC analysis, offering rapid and comprehensive information about the detected compounds.

Mobility spectrometry separates chemical components based on their ion mobilities within an electric field. It can be divided into ion mobility spectrometry (IMS) and field asymmetric ion mobility spectrometry (FAIMS) [34]. These methods often provide online analysis.

Over the past decade, advancements in electronic nose (e-nose) technology, which leverages nanosensors and pattern recognition algorithms, have garnered significant attention [35,36]. In this approach, a patient’s exhaled breath is directed through a mouthpiece into a sensor chamber containing an array of nanosensors capable of interacting with VOCs [37,38]. The output signals, representing the conductivity values of the sensor array, are then analyzed using machine learning models to generate a “breath print” that characterizes the overall VOC composition of the individual’s breath.

It is important to note that various environmental and physiological factors, such as diet, smoking, alcohol consumption, and ambient air quality, can significantly influence the VOC patterns in exhaled breath. Researchers have adopted several measures to minimize the impact of these confounding factors, including the use of carbon filters to eliminate environmental air contamination and high-efficiency particulate air (HEPA) filters to block bacteria and viruses from entering the sensor chamber [37]. In some studies, participants are asked to fast and abstain from smoking and alcohol prior to breath sampling, while in others, these variables are simply recorded as baseline information [39]. Although some research has attempted to examine the influence of factors like food intake on e-nose performance [38], a clear consensus has not yet been established, and no standardized sampling protocol has been widely adopted.

The simplicity of the sampling procedure and the application of nanosensor and machine learning technologies have greatly reduced the analysis time in e-nose systems. For instance, the Aeonose^TM^ e-nose product can complete the entire analytical process, from the start of one breath to the readiness of the next, in just 15 min [40]. Despite these obvious advantages in terms of operational simplicity, portability, rapid analysis, and cost-effectiveness, e-nose technology has not yet been widely adopted in clinical settings for tumor diagnosis [41]. This is largely due to the high variability in results observed across different e-nose devices, as well as the opacity of the analysis process and the underlying machine learning techniques, which can hinder the reproducibility of outcomes and the comparability of results produced by different machines or under varying environmental conditions. Additionally, unlike mass spectrometry, which can individually identify gases based on their mass-to-charge ratio, e-nose devices provide a collective analysis of VOCs [42].

### 3.3. Gastro-Esophageal Reflux Disease and Barrett’s Esophagus

BE is characterized by a change in the distal esophageal epithelium, identified as columnar-type mucosa during endoscopy, and confirmed to have intestinal metaplasia through biopsy. It is a known precancerous condition for esophageal adenocarcinoma, with a relative risk of 11.3 compared to the general population [43]. The prevalence of BE and the incidence of esophageal adenocarcinoma have been on the rise in Western countries, mainly attributed to the increasing prevalence of GERD, obesity, and a decline in Helicobacter pylori infection [44]. Similar trends have also emerged in Asian countries recently [45].

Current guidelines from professional societies recommend considering screening for BE in patients with multiple risk factors or the presence of GERD along with additional risk factors such as male sex, age over 50 years, White race, tobacco smoking, obesity, and family history of BE or esophageal adenocarcinoma [46,47,48]. However, a study involving primary care clinic patients found that over 50% of patients diagnosed with BE did not have frequent GERD symptoms [49]. This highlights the low sensitivity of guidelines that rely on GERD symptoms along with additional risk factors. Conversely, guidelines that do not require GERD symptoms have low specificity, leading to potential misdiagnosis [49]. It is reported that more than 50% of BE cases go undiagnosed in the community [50].

Endoscopy, the current gold standard for BE screening, has its limitations, including high procedural costs, time away from work, risks associated with sedation, and variable quality. Studies have shown that EAC can be missed during the initial endoscopy, leading to a significant percentage of post-endoscopy EAC diagnoses [51,52]. The gaps in current guidelines and referrals offer opportunities for developing and using new technologies to enhance the identification and screening of at-risk patients.

VOCs can represent various metabolic and disease states. E-nose, equipped with metal oxide sensors, has shown potential as a novel screening method for BE. By detecting disease-specific patterns of VOCs in exhaled breath, the e-nose can distinguish between patients with and without BE. In a study involving 401 patients, the e-nose demonstrated promising diagnostic accuracy, with a sensitivity of 91% and specificity of 74% [9]. Importantly, the diagnostic accuracy was not affected by a history of proton pump inhibitor use, GERD, or a hiatal hernia. Additionally, acetic acid in breath has been identified as a potential marker of gastro-esophageal reflux disease, as it leads to a decrease in the pH of the airway lining [11]. However, further validation of this technology in a primary care setting is necessary, along with a better understanding of factors that may influence VOCs, such as medications and diet.

### 3.4. Coeliac Disease

Coeliac disease is a chronic inflammatory condition of the small intestine triggered by the consumption of gluten in genetically susceptible individuals, along with potential involvement of other environmental factors [53]. The clinical presentation of coeliac disease can vary significantly, particularly depending on the age at which it is diagnosed. Patients may experience gastrointestinal symptoms, symptoms outside the gastrointestinal tract, or they may be asymptomatic [54]. This variability in symptoms can make the diagnosis challenging. While serological screening is widely employed and the response to a gluten-free diet is an important indicator, coeliac disease is ultimately diagnosed through the examination of duodenal biopsies. Multiple biopsies are typically required to establish a definitive diagnosis [55,56]. Therefore, the development of non-invasive or less invasive diagnostic methods would be highly beneficial for coeliac disease diagnosis, reducing the need for invasive procedures such as biopsies.

In a recent study, the researchers monitored the VOCs excreted in the exhaled breath of 20 healthy individuals over time. These individuals adhered to a gluten-free diet (GFD) for four weeks before switching to a normal diet. The study utilized gas chromatography-time-of-flight mass spectrometry (GC-ToF-MS) to analyze the VOCs. The authors identified a set of 12 volatile compounds, with 7 of them chemically identified that were able to differentiate between the GFD and a normal diet. However, the study did not mention the validation of the multivariate model used [57].

In a subsequent study, 33 participants were enrolled, including 16 patients with coeliac disease who were all adhering to a gluten-free diet, and 17 healthy controls. Interestingly, none of the previously identified VOCs showed significant differences between the two groups. The concentration range of the exhaled VOCs measured and identified in this study was consistent with findings in healthy individuals. Therefore, the results of this study suggest that the exhaled breath of patients with coeliac disease on a gluten-free diet is similar to that of healthy individuals [58].

The reason why the two studies showed inconsistency may be that the patients enrolled in the second research were on a GFD for at least a year, and they were without active symptoms of coeliac disease. Further investigations will be undertaken to verify whether breath analysis by PTR-MS can be used to test and monitor the nonadherence of CDP to the GFD or to identify non-invasive markers of coeliac disease and provide a non-invasive and rapid screening tool.

### 3.5. Inflammatory Bowel Disease and Irritable Bowel Syndrome

IBS is the most prevalent disorder of gut–brain interaction, characterized by cyclic abdominal pain and changes in bowel habits [59]. It affects approximately 4% to 10% of the global population [59]. The pathophysiology of IBS is not well understood but is believed to involve low-grade inflammation, altered gut microbiota, impaired mucosal barrier, and visceral hypersensitivity. Current diagnostic criteria, based on symptoms, pose challenges in clinical practice [60]. IBS patients are often subjected to multiple investigations tests such as fecal lactoferrin and calprotectin, and measurements of serum erythrocyte sedimentation rate (ESR) and C-reactive protein (CRP). Invasive procedures like colonoscopy are also commonly performed, with colonoscopy rates varying from 35% in the UK to 87% in France [61]. To address these challenges, there is an urgent need for noninvasive biomarkers, ideally in the form of a point-of-care test, that can accurately diagnose IBS and reduce the need for unnecessary procedures, specifically for patients who meet the ROME IV criteria.

Given that both IBD and IBS are connected to low-grade inflammation and alterations in the microbial ecosystem, VOCs present a non-intrusive tool that can assist in the diagnosis, treatment, and monitoring of these conditions. VOCs can be utilized to differentiate between various disease subtypes and phenotypes, as well as to determine the presence of active or inactive disease, assess treatment efficacy, and investigate the impact of the microbiota on VOC composition.

Several studies have suggested the potential role of breath VOCs as markers of inflammatory bowel disease. Raised concentrations of pentane, hexane, hydrogen sulfide, propionic acid, butanoic acid, and acetic acid are found in inflammatory bowel disease patients. These breath profiles may be associated with bacterial dysbiosis and lipid peroxidation caused by oxidative stress [62]. Nevertheless, it is important to note that individual VOCs exhibit limited specificity. While they are generally associated with inflammation, their levels can also be altered in other diseases. For instance, pentane, which has been found in both Crohn’s disease (CD) and ulcerative colitis (UC), has also been identified in asthma, heart failure, and non-alcoholic fatty liver disease [63]. The levels of breath pentane serve as an indicator for lipid peroxidation, suggesting its potential as a marker for general inflammation, ferroptosis, or apoptosis. Also, propanol has been previously observed in individuals with coeliac disease [58]. Consequently, it becomes crucial to combine these VOCs in discriminative models along with other clinical and biochemical data to enhance disease specificity, particularly in the context of monitoring. Furthermore, it is essential to incorporate appropriate control groups. Many of the studies included in this review compared patients with healthy controls. However, clinically, it would be more beneficial to focus on comparing symptomatic groups of patients. Therefore, future studies should concentrate on comparing these specific symptomatic groups.

Moreover, when it comes to accurate differentiation of IBS and IBD in the context of a differential diagnosis, it is crucial to not only distinguish them from each other but also from other gastrointestinal disorders. Therefore, research should ideally encompass a broader range of gastrointestinal disorders in a case–control design, allowing for result comparison and optimization of specificity. It is important to recognize that IBS and IBD encompass heterogeneous populations [64]. Consequently, pooling data is not recommended as it can distort results and overlook significant differences. For IBS, it is advisable to classify patients according to the Rome IV criteria, which include subtypes such as diarrhea-predominant, constipation-predominant, mixed, and unspecified. Since the underlying pathophysiology is presumed to differ among these subtypes, it is expected that distinct VOC patterns would emerge. Similarly, for IBD patients, it is essential to differentiate between CD and UC, and further subdivide them based on disease activity or remission status, as this could reveal valuable discriminatory characteristics. A proper sample size calculation should be employed to determine the total number of patients required to adequately examine subgroup characteristics.

### 3.6. Chronic Liver Disease

Chronic liver diseases (CLD) encompass a range of pathologies characterized by prolonged liver impairment and loss of function. Among the well-known types of CLD are hepatic encephalopathy, liver cirrhosis, and non-alcoholic fatty liver disease. These conditions have a profound impact globally, resulting in numerous direct (liver function loss) and indirect (secondary pathologies associated with CLD) deaths each year. Furthermore, their incidence is steadily increasing, posing a significant and escalating burden worldwide [65,66].

#### 3.6.1. Cirrhosis

Liver cirrhosis (LC), also referred to as alcoholic hepatitis, primarily arises from high-risk behaviors, notably excessive alcohol consumption. This pathology is characterized by the progressive loss of hepatic functions and, in severe cases, can lead to the development of hepatic carcinoma. The conventional diagnostic approach for liver cirrhosis, liver biopsy, is an invasive procedure associated with potential complications [67,68,69].

The identification of VOCs in exhaled breath has emerged as a promising diagnostic method for liver cirrhosis. Dadamio et al. (2012) investigated 23 VOCs as potential biomarkers for LC diagnosis. Exhaled breath samples from LC patients and healthy individuals were analyzed using GC-MS. By employing a distinct VOC pattern, the authors achieved differentiation between the two groups with sensitivity and selectivity levels ranging from 82% to 88% and from 96% to 100%, respectively [29]. Similarly, Morisco et al. (2013) identified a group of VOCs exhibiting substantial potential as biomarkers for LC diagnosis. The results obtained through PTR-ToF-MS measurements represent the first-ever “breath-print” of liver cirrhosis, according to the authors [28].

In cases of alcohol-related cirrhosis, Hanouneh et al. (2014) analyzed exhaled breath samples from a cohort of 123 volunteers, including 43 healthy individuals, 40 alcoholic cirrhosis patients, and 40 non-alcoholic cirrhosis patients. Using a mass spectrometer, the authors identified six VOCs—trimethylamine, pentane, ethanol, acetone, acetaldehyde, and isopropanol—that exhibited increased levels in LC patients compared to healthy individuals [27].

In the first attempt to distinguish compensated cirrhosis (precursor condition of decompensated cirrhosis with clinically evident complications like ascites, variceal hemorrhage, hepatic encephalopathy, and jaundice) from a heterogeneous group of CLD patients using VOCs, a set of 11 volatiles outperformed a combination of 5 serum markers (GGT, ALT, bilirubin, albumin, and thrombocytes). The VOC test demonstrated higher sensitivity (0.83) and specificity (0.87) compared to the serum markers test. Interestingly, combining both approaches did not improve accuracy [24]. This study is notable for including CLD patients without cirrhosis as a reference group, emphasizing the importance of early diagnosis to prevent disease progression and complications. While liver biopsy is the current standard, its invasiveness and associated morbidity call for less invasive alternatives.

Factors influencing the course of cirrhosis are extremely variable, and many prognostic models and scores have been proposed in the last decades. The Child–Pugh Classification (CPC) [70,71] is simple to perform at the bedside but relies on subjective variables and may not consider other important risk factors. Additionally, the cut-off values for laboratory variables may not be optimal, and the explained survival variation is relatively low [72]. Attempts to improve the CPC with additional variables have shown limited success. The model for end-stage liver disease (MELD) score [73], initially promising for prognostic prediction, did not consistently outperform the CPC in terms of explained survival variation. Therefore, comprehensive prognosis estimation for cirrhosis patients remains challenging. In a pilot study, BPs identified by e-nose were found to be associated with significant clinical endpoints such as mortality and hospitalization, independent of “classical” prognostic indices. The use of e-nose provides the advantages of cost-effectiveness and speed. On the downside, it does not allow for the identification of individual VOCs or the interpretation of blood pressure, which hinders the authors from establishing clinical correlations.

#### 3.6.2. Hepatic Encephalopathy

Hepatic encephalopathy (HE) is a condition that occurs as a result of chronic liver diseases when the liver’s impaired function prevents the elimination of harmful substances from the body, leading to their accumulation and causing neurological dysfunction. Symptoms of HE include disorientation, dizziness, changes in mood and behavior, alterations in sleep patterns, and even coma [74]. To diagnose hepatic encephalopathy in patients with chronic liver disease more quickly and accurately, Khalid et al. (2013) conducted a study to evaluate the use of breath patterns to differentiate between different groups of patients: alcoholic cirrhosis patients with hepatic encephalopathy (11 volunteers), alcoholic cirrhosis patients without hepatic encephalopathy (23 volunteers), non-alcoholic cirrhosis patients without hepatic encephalopathy (13 volunteers), harmful drinkers without cirrhosis (7 volunteers), and 15 healthy volunteers. They analyzed breath samples (1 L) using a GC-MS device and successfully identified hepatic encephalopathy in 90.9% of the cases by analyzing the breath patterns. Furthermore, the breath patterns also allowed for the accurate identification of liver cirrhosis with 100% accuracy [75]. Arasaradnam et al. (2016) also focused on diagnosing hepatic encephalopathy through exhaled VOCs. They used an analytical technique called ultraviolet-field asymmetric ion mobility spectrometry (UV-FAIMS) to analyze breath samples collected from 42 volunteers, including 20 healthy individuals and 22 hepatic encephalopathy patients. By examining the respective VOC patterns, the authors were able to distinguish between the two groups with a specificity level of 68% and a sensitivity level of 88% [76]. O’Hara et al. (2016) identified limonene as a particularly interesting biomarker in the study of hepatic encephalopathy. They collected and analyzed breath samples from 61 volunteers, including 31 hepatic encephalopathy patients and 30 healthy individuals, using a proton transfer reaction-quadrupole-mass spectrometry (PTR-Quad-MS) device. Among all the compounds detected, the authors found that limonene had significantly lower concentrations in hepatic encephalopathy patients compared to healthy individuals, making it a potential biomarker [77].

#### 3.6.3. Non-Alcoholic Fatty Liver Disease

Non-alcoholic fatty liver disease (NAFLD) is the leading cause of CLD worldwide and is responsible for a significant number of liver transplant cases. Unlike alcoholic liver disease, NAFLD is primarily associated with risk factors such as overweightedness, obesity, and unhealthy lifestyles [78]. While some studies have focused on analyzing VOCs in fecal and urinary samples, Han et al. (2019) developed a micro gas chromatography (GC) column specifically for studying NAFLD biomarkers in the exhaled breath of CLD patients, demonstrating its potential for NAFLD diagnosis [79].

Verdam et al. investigated the use of breath analysis as a non-invasive and accurate method for diagnosing NAFLD. They used GC-MS to detect exhaled analytes from 65 individuals, including 26 healthy individuals and 39 NAFLD patients. Among the compounds detected, the authors identified four specific VOCs (3-methylbutanonitrile, propanol, α-terpinene, and tridecane) that allowed them to distinguish between the two groups with an accuracy of 77%, a negative predictive value of 82%, and a positive predictive value of 81% [80]. Additionally, seven volatile organic compounds (dimethyl sulphide, limonene, acetophenone, α-terpinene, isoprene, acetone, and styrene) were found to be of particular interest as potential NAFLD biomarkers. To evaluate their significance, the authors analyzed the exhaled breath of 43 volunteers, including 14 healthy subjects, 15 cirrhotic NAFLD patients, and 14 non-cirrhotic NAFLD patients. Using these seven VOCs, they were able to differentiate between healthy volunteers and NAFLD patients with an accuracy level of 95%. Furthermore, they achieved a 95% accuracy in distinguishing between cirrhotic and non-cirrhotic NAFLD patients.

### 3.7. Progressing VOC Breath Tests toward Clinical Application

The typical path for developing biomarker-based diagnostics involves an unbiased discovery phase followed by targeted validation—both analytical and clinical [81]. In this review, we examine the promising progress in VOC-based breath tests for non-cancer gastrointestinal diseases. These tests have demonstrated exciting advancements in early detection, prognosis, and disease monitoring.

Clinical validation remains a critical yet challenging hurdle. As the field advances, the focus is shifting to analytical validation. While initial studies have shown VOC breath tests can meet crucial clinical needs, few biomarkers have been clinically validated thus far. A key limitation has been the lack of standardization in study design, sample collection, analysis, and reporting [62].

However, recent innovations have improved the quality and consistency of breath studies, with commercial tools now available. We anticipate a surge in validation studies replicating and expanding on existing research. Crucially, establishing robust biological evidence linking specific VOCs to disease processes will be essential to justify clinical use of these biomarkers. For example, Ratcliffe et al. recently tracked how unsaturated fatty acids are broken down into certain alkanes as a result of lipid peroxidation [82]. Furthermore, methane and hydrogen have long been validated as products of intestinal anaerobic bacteria. This could include insights into how certain metabolites are produced during lipid peroxidation or microbial activities [83].

Standardizing result reporting and understanding healthy VOC variation will also be pivotal. The breath tests closest to clinical implementation leverage well-characterized biomarkers from administered substrates, drawing on prior biological knowledge. Novel EVOC probes to assess diverse metabolic pathways could further expand the applications of this promising diagnostic approach [84].

## 4. Conclusions

In conclusion, this narrative review of recent literature highlights the potential of breath analysis as a noninvasive method for diagnosing and monitoring non-cancer gastrointestinal diseases. Metabolic and inflammatory conditions have been shown to be associated with distinct respiratory profiles [85].

However, despite the significant advancements in this field, VOCs have yet to be established as clinically useful and validated biomarkers for any specific disease. Technical and quality issues have hindered the development of reliable biomarker tools. While potential biomarkers have demonstrated statistical significance and diagnostic models have shown promising sensitivity, specificity, and area under the curve (AUC) values, it remains challenging to demonstrate the disease specificity of these VOCs in exhaled breath, particularly in advanced disease stages where overlapping patterns may exist with other conditions. Recent studies have highlighted the potential of exhaled breath testing for cancer diagnosis, particularly in gastrointestinal cancers such as esophageal, gastric [86], and colorectal cancers [86,87,88]. Additionally, there is a growing body of research on the use of volatile organic compound (VOC) breath tests for the detection of hepatocellular carcinoma [3,4]. These studies have demonstrated several advantages of VOC breath tests over conventional diagnostic methods, including comparable sensitivity, simplicity, and cost-effectiveness. However, a key limitation of current VOC breath test research is the focus on discriminating between patients with advanced cancer and healthy controls [89]. This approach fails to account for the fact that patients with various advanced diseases or conditions are likely to exhibit elevated levels of inflammatory or oxidative stress-related VOCs, regardless of the specific disease etiology. As a result, the specificity of VOC breath tests in the present study design may not be reliable, as the underlying differences in VOC metabolism between cancer patients and those with other chronic diseases are not adequately explored. To address this issue, future research on VOC breath tests in cancer diagnosis should emphasize the ability to distinguish between advanced cancer conditions and their precursor lesions. For instance, studies on BE and esophageal–gastric junction cancer have revealed overlapping biomarkers [90,91,92], suggesting the need for further research comparing VOC profiles of patients with BE-related symptoms to those with cancer. This approach would not only improve the specificity of VOC breath tests but also contribute to our understanding of the underlying metabolic pathways involved in carcinogenesis. Furthermore, there is a need to refine the sampling procedures employed in these studies. Collecting breath samples at a single time point may not be sufficient to capture the dynamic nature of VOCs. Additionally, the composition of VOCs can be influenced by the surrounding air, known as the exposome, which necessitates the collection of background samples and the correction for external influences. Detailed descriptions of sample collection, handling, storage conditions, and preparation should be provided by the authors.

In future studies, a deeper understanding of the metabolic pathways underlying VOCs could shed light on the pathophysiological mechanisms of these diseases. Further analysis of these pathways and VOCs of interest in in vivo models and animal research may lead to the identification and development of novel therapeutic targets. For instance, short-chain fatty acids have emerged as a notable group of compounds in studies of IBD and IBS, given their established role in the inflammation pathway. Additionally, comparing the VOC composition in breath, urine, and feces from the same patient could provide insights into the metabolic processes involved in the disease and elucidate VOC metabolism from the gut to the breath.

## Figures and Tables

**Figure 1 biomedicines-12-01815-f001:**
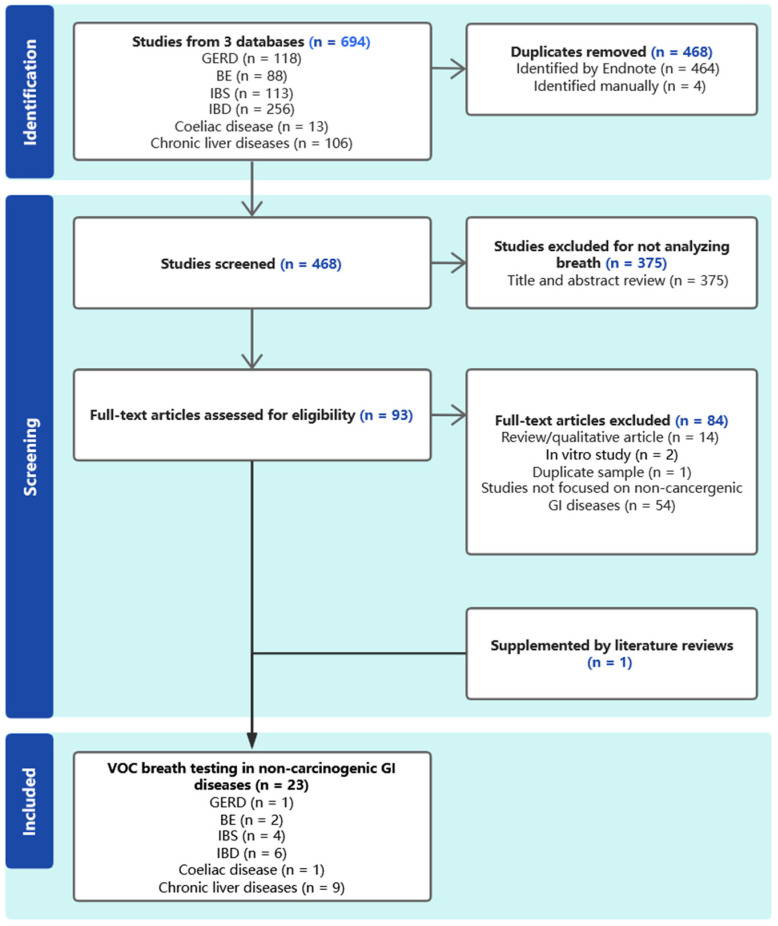
PRISMA flowchart describing literature search strategy.

**Figure 2 biomedicines-12-01815-f002:**
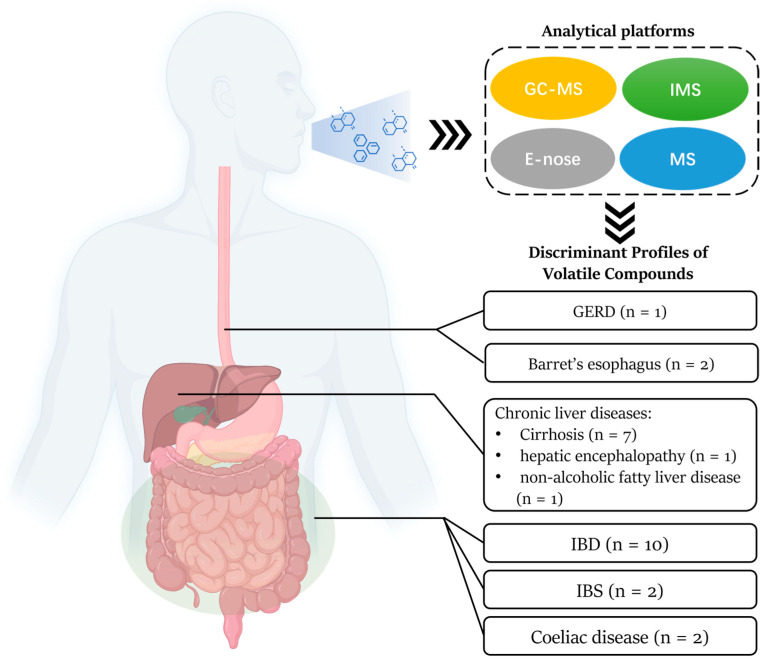
Analytical platforms and numbers of studies on VOC breath testing for non-cancer gastrointestinal disorders.

**Figure 3 biomedicines-12-01815-f003:**
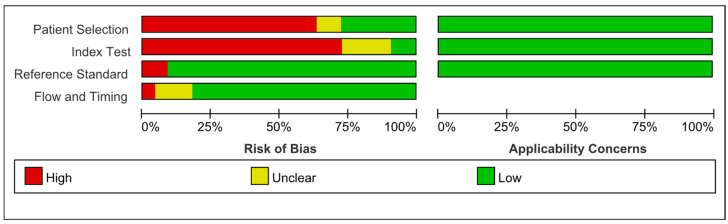
Quality assessment of the studies of risk of bias and concerns regarding applicability based on QUADAS 2.

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
