# Peer review of "Prospect and Challenges of Volatile Organic Compound Breath Testing in Non-Cancer Gastrointestinal Disorders"

_biomedicines, 2024, doi:10.3390/biomedicines12081815_

Round 1

Reviewer 1 Report

Comments and Suggestions for Authors

The manuscript is not novel; it is just a review of well-known data. AI tools can now automatically generate it.

Author Response

Thank you very much for taking the time to review this manuscript.

We are sorry to hear that you feel this manuscript lacks novelty. While it is indeed a review article and includes a significant portion of well-known data in the field of VOC breath testing in non-cancer gastrointestinal diseases, we must disagree with the point that AI tools could generate this article. Although large language models are becoming increasingly capable of analyzing and integrating published information, they lack the ability to raise questions from clinical practice, identify gaps in current studies, and guide future research developments.

Through this review, we raise an important question: why, despite the long history of breath tests dating back to 460 BC and the new analytical platforms such as e-noses that allow for the analysis of a VOC profile within 15 minutes, there is still limited clinical application? Although there is a growing body of research in GI cancer, we chose to focus on non-cancer diseases because a key limitation of current VOC breath test research is its focus on differentiating between patients with advanced cancer and healthy controls. This approach overlooks the fact that patients with various advanced diseases may exhibit elevated levels of inflammatory or oxidative stress-related VOCs. Future research should emphasize distinguishing between advanced cancer conditions and their precursor lesions to improve the specificity of VOC breath tests.

Moreover, disease diagnosis by VOCs is much more complex than diagnosis by blood tests, which often involve one chemical being strongly associated with one disease. Numerous studies aiming to find a specific VOC for a disease have failed. Therefore, we suggest that future studies focus on understanding the metabolic pathways underlying VOCs to shed light on the pathophysiological mechanisms of these diseases rather than searching for a single VOC as a biomarker. The points we raise above have not been mentioned in existing review articles, despite the many reviews in this field.

Additionally, we have revised the article according to feedback from other reviewers. After a systematic literature search conducted on July 1st through EMBASE, MEDLINE, and Cochrane using the terms mentioned in the Method section and applying the inclusion and exclusion criteria, we updated the list of studies in Table 1 and evaluated their quality. We believe this article is valuable for researchers in the field of VOC analysis and for clinicians interested in non-invasive diagnostic tests and surveillance.

Thank you again for your review and suggestions.

Reviewer 2 Report

Comments and Suggestions for Authors

The paper examines an important deficiency in the existing diagnostic field by specifically investigating the potential of volatile organic compounds (VOCs) in breath analysis for diseases other than cancer. Prior to publishing of the essay, the authors must take into account the following factors:

1. The methodology lacks a vital element that is essential for any review study, namely the inclusion and exclusion criteria. 2. Another crucial factor is the authors' method of assessing the quality of the selected papers and the methods they utilized for this purpose. 3. The references and studies in Table 1 are very outdated. It needs updating using the most current state-of-the-art methods. 4. The abstract states that the clinical validation of VOCs as biomarkers for specific diseases has not yet been achieved. The absence of clinical validation presents a notable obstacle to the prompt use of VOC breath testing in clinical environments. The review would be enhanced by an analysis of the necessary procedures for clinical validation and the possible obstacles encountered in this undertaking. 5. The review's objective is to investigate non-cancer disorders. However, the decision to exclude cancer-related studies may restrict the generalizability of the findings. Considering the extensive research conducted on volatile organic compounds (VOCs) in cancer diagnosis, incorporating a comparison or reference to this field could offer a more comprehensive perspective on the study of VOCs in breath. Comments on the Quality of English Language

Minor editing of English language required.

Author Response

Dear Reviewer,

Thank you very much for your valuable advice. As post-graduate students, we greatly value your opinion and are more than willing to revise the article accordingly. Here's our point-to-point response:

Comments 1: The methodology lacks a vital element that is essential for any review study, namely the inclusion and exclusion criteria.
Response 1: Thank you. We've now made it clear in the Method section (line 96-115), and we visually display the selection process via PRISMA flow chart (Figure 1).

Comments 2: Another crucial factor is the authors' method of assessing the quality of the selected papers and the methods they utilized for this purpose.
Response 2: Thank you. We now assess the quality of each study included was assessed using the quality assessment tool for diagnostic accuracy studies (QUADAS-2). And the results are shown in 3.1 Quality assessment of studies (line 126-146).

Comments 3: The references and studies in Table 1 are very outdated. It needs updating using the most current state-of-the-art methods.
Response 3: Thank you. After systematic search on 1st July of studies through EMBASE, MEDLINE and Cochrane using combinations of the terms mentioned in the Method section and adopted the inclusion and exclusion standards, we updated the list of research in Table 1. However, studies regarding VOC breath tests in non-cancer GI diseases are relatively scarce and there's no studies published in 2024.

Comments 4: The abstract states that the clinical validation of VOCs as biomarkers for specific diseases has not yet been achieved. The absence of clinical validation presents a notable obstacle to the prompt use of VOC breath testing in clinical environments. The review would be enhanced by an analysis of the necessary procedures for clinical validation and the possible obstacles encountered in this undertaking.
Response 4: Thank you very much. We find it a wonderful idea and we decided to write a independent section on how to process VOC breaths tests into clinics (3.7 line 442-466).

Comments 5: The review's objective is to investigate non-cancer disorders. However, the decision to exclude cancer-related studies may restrict the generalizability of the findings. Considering the extensive research conducted on volatile organic compounds (VOCs) in cancer diagnosis, incorporating a comparison or reference to this field could offer a more comprehensive perspective on the study of VOCs in breath.
Response 5: Thank you. We can't agree more that there's a ever-growing number of VOC breath tests in cancer diagnosis. However, we found the focusing on discriminating between patients with advanced cancer and healthy controls, rather than exploring the underlying differences in VOC metabolism between cancer patients and those with other chronic diseases limits the specificity of VOC breath tests, as patients with various advanced diseases may exhibit elevated levels of inflammatory or oxidative stress-related VOCs. Also we would like to see more studies regarding non-cancer GI diseases, which often receive less attention than necessary in diagnostic research. Therefore, we decided to orient this research on non-cancer diseases, although there're fewer studies. We further discuss the comparison between cancer and non-cancer in the discussion section (Line 472-499).

We thank you again for your thoughtful review and recommendations, which have helped us strengthen the article. Please let us know if you have any other feedback or suggestions.

Sincerely,

Ke Pang

Reviewer 3 Report

Comments and Suggestions for Authors

Review report for biomedicines-3085192 entitled “Prospect and Challenges of Volatile Organic Compound Breath Testing in Non-cancer Gastrointestinal Disorders“

This review article focused the evaluating current proof-of-concept study of breath-borne volatile organic compounds (VOCs) to screen or diagnose diseases. The reviewer agreed the conclusions of the manuscript: "In future studies, a deeper understanding of the metabolic pathways underlying VOCs could shed light on the pathophysiological mechanisms of these diseases.“ Disease diagnosis by VOCs is much more complex than diagnosis by blood tests (often one chemical is strongly associated with one disease), and no amount of review articles summarizing the research to date will suffice. The reviewer suggest to the editor accepting this manuscript after the minor revision.

Below are comments regarding areas for improvement.

1. Since the abbreviation definitions for some words are given in multiple places, please leave the first one alone.

2. In section3, GC-MS are explained twice.

3. There are various methods for e-nose. e-nose is defined as a device that classifies gas mixtures from pattern recognition output by a sensor array containing multiple low-selectivity sensors. It is called e-nose because the above classification scheme is modeled after the olfactory function of living organisms. I believe the notation in the manuscript is incorrect and needs to be investigated and re-written.

Author Response

Dear Reviewer,

Thank you very much for your valuable advice. As post-graduate students, we greatly value your opinion and are more than willing to revise the article accordingly. Here's our point-to-point response:

Comments 1: Since the abbreviation definitions for some words are given in multiple places, please leave the first one alone.
Response 1: Thank you very much. We've checked the abbreviations and repetition is eliminated.

Comments 2: In section3, GC-MS are explained twice.
Response 2:Thank you again, we now removed it.

Comments 3: There are various methods for e-nose. e-nose is defined as a device that classifies gas mixtures from pattern recognition output by a sensor array containing multiple low-selectivity sensors. It is called e-nose because the above classification scheme is modeled after the olfactory function of living organisms. I believe the notation in the manuscript is incorrect and needs to be investigated and re-written.
Response 3: Thank you for pointing this out, we're rewritten the description of e-nose in section 3.2 (line 178-208).

We appreciate you taking the time to provide this constructive feedback. The revisions you suggested have helped us improve the clarity and consistency of the manuscript. Please let us know if you have any other comments or suggestions.

Sincerely,

Ke Pang

Round 2

Reviewer 2 Report

Comments and Suggestions for Authors

This manuscript can be accepted by Biomedicines.